# X under Musk's leadership: Substantial hate and no reduction in inauthentic activity

**Daniel Hickey**[1], **Daniel M. T. Fessler**[2], **Kristina Lerman**[3], **Keith Burghardt**[3¤]*

**1** School of Information, University of California, Berkeley, Berkeley, California, United States of America, **2** Department of Anthropology and UCLA Bedari Kindness Institute, University of California, Los Angeles, Los Angeles, California, United States of America, **3** USC Information Sciences Institute, Marina del Rey, California, United States of America

¤ Current address: IMDb.com, Inc., Seattle, WA, USA
* keithab@isi.edu

**Data Availability Statement:** All data IDs and code are accessible from the following link: https://github.com/dan-hickey1/x-under-musks-leadership.

## Abstract

Numerous studies have reported an increase in hate speech on X (formerly Twitter) in the months immediately following Elon Musk's acquisition of the platform on October 27th, 2022; relatedly, despite Musk's pledge to "defeat the spam bots," a recent study reported no substantial change in the concentration of inauthentic accounts. However, it is not known whether any of these trends endured. We address this by examining material posted on X from the beginning of 2022 through June 2023, the period that includes Musk's full tenure as CEO. We find that the increase in hate speech just before Musk bought X persisted until at least May of 2023, with the weekly rate of hate speech being approximately 50% higher than the months preceding his purchase, although this increase cannot be directly attributed to any policy at X. The increase is seen across multiple dimensions of hate, including racism, homophobia, and transphobia. Moreover, there is a doubling of hate post "likes," indicating increased engagement with hate posts. In addition to measuring hate speech, we also measure the presence of inauthentic accounts on the platform; these accounts are often used in spam and malicious information campaigns. We find no reduction (and a possible increase) in activity by these users after Musk purchased X, which could point to further negative outcomes, such as the potential for scams, interference in elections, or harm to public health campaigns. Overall, the long-term increase in hate speech, and the prevalence of potentially inauthentic accounts, are concerning, as these factors can undermine safe and democratic online environments, and increase the risk of offline harms.

## Introduction

*"If our Twitter bid succeeds, we will defeat the spam bots or die trying!"* (11:53 AM PDT, April 21, 2022)

*"Hate speech impressions (# of times tweet was viewed) continue to decline, despite significant user growth!"* (10:53 AM PDT, December 2, 2022)

–Elon Musk

**Funding:** The full name of each funder DH is funded through the National Science Foundation (award #2051101; https://www.nsf.gov/), who did not play any role in the study design, data collection and analysis, decision to publish, or preparation of the manuscript. KL and KB are funded through the Defense Advanced Research Projects Agency (awards #HR0011260595 and #HR001121C0169; https://www.darpa.mil/), who did not play any role in the study design, data collection and analysis, decision to publish, or preparation of the manuscript.

**Competing interests:** This work was done prior to author Keith Burghardt joining Amazon and IMDb.

Before Elon Musk officially purchased Twitter (now known as X) and became its CEO on October 27th, 2022, he publicly made several promises as to how he would change the platform, including fewer restrictions on content moderation and a greater crack down on automated spam accounts on the platform [1]. Since his takeover, many changes have been made in the internal operations of the company, as well as in the user experience on the platform. Immediately following the purchase, there was a reshuffling of leadership, and the majority of workers, including many within the trust and safety team, left the company through resignations and layoffs [2–4]. The Trust and Safety Advisory Council, which advised on content moderation, was disbanded [5]. Additionally, the demographics of X likely shifted following Musk's purchase, with an observed decrease in activity for many environmental activists [6] and many users migrating to Mastodon [7], Threads [8], and BlueSky [9].

While Musk's promises to reduce bots and inauthentic accounts have been lauded, actions taken by Musk to reduce other types of content moderation on X have been met with concern [10–13]. Platforms generally moderate content that is harmful to other users, such as hate speech [14]. A rise in hate speech on a mainstream social media platform such as X is troubling for a number of reasons. The prevalence of online hate is linked to offline hate crimes [15], and victims of hate often report decreased psychological well-being [16–18]. Additionally, exposure to hate ideologies can increase prejudice [19] and decrease empathy towards outgroups [20]. Given that many of the aforementioned harms of hate speech are related to exposure to such speech, simply measuring the volume of hate speech will not address whether these harms have been amplified, as it is possible more posts are produced, but they are seen by fewer people. Publicly, X has taken the stance of "freedom of speech, not freedom of reach," stating that, while the platform will take a looser approach to policing hate speech, it will not show hate speech to many users [21]. Musk and X have claimed that overall exposure to hateful content and spam on X has decreased since Musk's acquisition [22, 23]. However, these claims lack transparency, and so must be evaluated independently. For example, X could have a fundamentally different definition of hate speech than the broader research community, as they now consider 'cisgender' a slur, despite this being a term widely used by those in medical [24] and gender nonconformity communities [25]. For this reason, we ask two research questions:

RQ1 How did Musk's purchase of X, and subsequent policy changes, correlate with the volume of hate speech on X?

RQ2 How did the level of engagement with posts containing hate speech change on X following Musk's purchase of the platform?

We use the term "engagement" to refer to the number of likes and reposts received by a given post on X. We are primarily interested in users' exposure to hate speech on the platform, yet we do not have data on the number of views each post received before Musk's purchase. Engagement metrics can therefore act as a reasonable proxy for the visibility of posts on X. While we cannot be certain that the number of likes received by a certain post directly corresponds to the number of views received by that post, we nevertheless find that these two metrics are strongly correlated. Previous independent reports show that hate speech has increased since Musk purchased X [26–29]; however, this prior research did not explore changes in the prevalence of hate speech over the longer term, and did not explore whether an increase in hate posts corresponds to an increase in engagement with hate.

In addition, and complementary to our study of hate, bots and information operations are a constant concern on social media platforms [30, 31], and have been used by governments and other actors to influence voters [32], manipulate markets [33], and generate other harms [34]. Information operations on social media, like bots, often involve the creation of fake accounts

[35] and the coordination of these accounts to push an ideology or disseminate false or misleading information [36]. It is therefore critical to understand the degree to which X has managed to affect the prevalence of inauthentic account activity, especially during the period following Musk's purchase [26], as his claim to have reduced social bots does not necessarily correspond to a decrease in such activity [37]. We therefore ask two additional research questions:

RQ3 How did Musk's purchase of X, and subsequent policy changes, correlate with inauthentic account activity on X?

RQ4 How did engagement with posts made by inauthentic accounts change following Musk's purchase of X?

## Our contribution

Given that previous analyses were conducted within weeks of Musk's purchase [26, 29], a great deal is unknown about how X has changed with regard to both hate and influence campaigns. Likewise, given that, as its owner, Musk undoubtedly can shape the platform regardless of his formal role, it is important to know whether changes to X observed by researchers immediately following Musk's purchase represent a short burst or a lasting trend in the platform during his tenure as CEO. To address these questions, we measure levels of hate speech and activity of inauthentic accounts on X from the beginning of 2022 to June 9th, 2023, the period representing Musk's entire tenure as CEO [38]. Using a lexicon of hate speech [26] and Google's Perspective API [39], we collected a set of posts that likely contain English-language hate speech, as well as corresponding engagement metrics for these posts. We evaluate changes in hate speech against a baseline sample of posts that were collected using common English words, finding that the total weekly volume of posts containing hate speech increased, with the post-purchase volume of posts consistently about 50% higher than the pre-purchase volume, whereas overall activity only increased by 8%. The weekly rate at which hate content was liked also significantly increased—by 70%—in contradiction to Musk's claims about decreased engagement with hate material (and in contrast to a modest 4% increase in likes within the baseline sample).

Equally troubling, we found no decrease, and a potential increase, in the activity of inauthentic accounts, such as bots. More specifically, we find that in the months after Musk bought X post activity from coordinated accounts did not change significantly, nor was there a reduction in attention to inauthentic posts via likes or reposts, based on three different account-coordination metrics. We also see a statistically significant increase in the bot scores of randomly sampled accounts after Musk's purchase, and a large upswing in posts promoting cryptocurrency (which have more bot-like behavior than random accounts). We acknowledge that, because we are not privy to specific internal changes in procedures, personnel, or priorities within X under Musk's tenure, we are unable to assert definitive causal claims regarding the patterns that we document. Moreover, we only explore English-language posts, thus changes in posts in other languages could be different. Subject to those limitations, in the concluding section of the paper we present plausible reasons for the observed results, and discuss their implications for X moving forward. Our contributions can be summarized as follows:

1. We track various dimensions of hate speech before and after Musk purchased X, finding significant increases in hate posts and engagement with these posts.

2. We track inauthentic accounts before and after Musk purchased X, finding no decrease, and a potential increase, in inauthentic account activity.

Overall, these results highlight a need for increased moderation to combat hate and inauthentic accounts on X.

## Related work

### Hate speech detection

While there is no universally agreed-upon definition of hate speech, hate speech detection is generally understood as the task of identifying whether a piece of text contains attacks or offensive language that targets specific identity groups (e.g., people of certain races, gender identities, sexual orientations, etc.) [40]. Defining hate speech remains a challenge in hate speech detection, as reliability is often low in datasets with annotated examples of hate speech [41]. Deep learning methods often achieve high performance on the task of automated hate speech detection [42, 43]. However, these methods lack explainability, meaning it is often difficult to understand why a given model classifies text as hate speech [44, 45]. Researchers have addressed this by incorporating human rationales into training datasets for hate speech detection [46]. Bias is another problem in hate speech detection, as models often erroneously interpret dialects of minority communities or benign terms as indicating hate speech [45, 47–49]. These biases are similarly seen in content moderation, as members of minority-dialect communities often report disproportionate rates of censorship on social media platforms [50]. While hate speech detection of high-resource languages, such as English, is commonly researched, there is also active research in improving hate speech detection for low-resource languages and dialects [51, 52]. Relatedly, the detection of hate is expanding to other modalities beyond text, including images and videos [53, 54].

### Content moderation and hate speech on social media

Social media platforms have employed many strategies to moderate their content. Platforms often moderate to reduce users' exposure to content such as hate speech or misinformation, and the most common form of moderation is the removal of posts containing these types of speech or accounts that often spread them [14]. An analysis of the deplatforming of prominent users who regularly posted offensive speech on X found that their followers posted less toxic content afterward [55]. Similarly, methods to quickly moderate on X may reduce harmful posts [56]. "Soft-moderation" techniques are also used, which involve placing warning labels on posts to alert users that they may contain misinformation [57–59]. On the platform Reddit, similar interventions are imposed on whole fora (called "subreddits"). One analysis of the banning of hate communities on Reddit found that overall hate speech on the platform decreased as a result [60]. Reddit also imposes soft-moderation on communities, called "quarantines," which warn users that the content of the community they are about to view contains potentially offensive material. Quarantines have been shown to reduce activity within problematic communities [61]. In addition to strategies imposed by platforms, "grassroots" approaches to moderation exist, where users attempt to stop the spread and impact of harmful content on a platform. One of the most prominent forms of this, known as "counterspeech," (speech that actively denounces hate speech), has been shown to reduce hate on X [62–64]. X also crowdsources from users rebuttals to misinformation, displaying rebuttals on posts that receive them with their "community notes" feature [65].

Content moderation decisions made by one social media platform may affect other platforms, as users who are deplatformed on one may migrate to another, spreading harmful content there instead [66–69]. Comparative studies of social media platforms find that those with high levels of moderation contain less hate speech [70].

## Account coordination

Information operations are an influence strategy often carried out on social media to promulgate propaganda and disinformation, with the goal of pushing ideologies [32, 34, 36, 71], sowing discontent [72], or driving pump-and-dump stock market scams [33], among others. While these operations often utilize social bots [31, 36], they can also be assisted by humans (*cyborgs*) or utilize accounts driven entirely by humans [73, 74]. There is often a strong overlap between the presence of bots and patterns of coordination [75, 76] (although a study of coordinated activity during the 2019 UK general election found relatively little correlation between bots and coordination [37]). These inauthentic accounts behave similarly with a common goal; such groups of accounts are known as drivers [77] or coordinated accounts [32, 71].

There are, however, a number of ways drivers may coordinate, such as pushing the same message, increasing the attention given to a post (even a post by an authentic account), or pushing similar (or dissimilar) messages at nearly the same time [33, 71]. The former takes advantage of humans' tendency to believe content that they encounter many times [32], or adopt ideas advocated by seemingly different people [78]. The reposting of a post, meanwhile, may make content appear more popular, increasing the likelihood of adoption (the bandwagon effect [79]). Finally, activity coordination could be useful in time-sensitive information campaigns, such as pump-and-dump schemes, or during a quickly-evolving situation, where a common message can drive many authentic users to take some action (such as buying a cryptocurrency or casting votes in an election).

While drivers need not always behave similarly to one another, highly similar accounts are a strong signal of a driver [71]. To this end, many researchers have explored how drivers can be detected based on post similarity [32, 80, 81], suspiciously fast interactions [71, 80, 81], or co-sharing of many URLs [82, 83], reposts [32, 33, 37, 71], or synchronous activity [32, 33, 71, 84]. Finally, previous researchers have explored latent representations of behavior [71, 85–89], such as embedding of activity [71, 77]. These or similar embeddings have been used successfully to detect social bots [90] that promote misinformation and propaganda (similar to coordinated accounts).

In contrast to previous work, we aim to study a broad sample of X data in the months before and after Musk purchased the platform, examining how his leadership has changed critical aspects of X. Specifically, we study how the behavior of English-language hate users has changed, and the activity of inauthentic accounts, both of which lead us to consider ways in which moderation can improve X.

# Methods

## Collection of posts containing hate speech

To identify posts containing hate speech, we follow methodology developed in previously published research devoted to tracking the volume of hate speech shortly after Musk's acquisition of X. Namely, we use a two-step process described by Hickey et al. [26], which consists of collecting an initial sample of posts that contain terms likely to be used in hate speech, then filtering out low-toxicity or highly sexual posts that are less likely to be associated with hate. For the first step, we collect posts containing at least one hate term as listed by Hickey et al., [26]. However, we remove the intellect-deprecating term "r****d" from Hickey et al.'s hate lexicon, as inspection indicates that this term–quite common in our dataset–often occurred in posts that, while hostile, did not directly target an identity group. Our results are robust to this change. The posts were collected at five randomly selected five-minute intervals per day using the Twitter API for Academic Research (now the X API). We then use the Perspective API [39] to

remove posts with toxicity confidence below 0.7 and with sexually explicit confidence above 0.3.

These choices are made to extract inflammatory content, as the "toxicity" model detects whether a given comment is "a rude, disrespectful, or unreasonable comment that is likely to make people leave a discussion." This model has been validated on X multiple times [91, 92]. Perspective also provides an "identity attack" model meant to detect posts that negatively target certain identity groups; we perform additional analyses using this model as a filter (with the same 0.7 threshold), finding our results to largely be the same. When considering posts that are included in our dataset, we opt to include reposts as well as original posts, with reposts of the same post being counted as many times as they appear in the dataset—our rationale being that a post's influence on the platform is amplified when it is shared many times, thus the increased representation of any given post in the dataset generated by the inclusion of reposts is appropriate. However, we recognize that if a repost is erroneously identified as hate speech, then its inclusion may impact the reliability of our results. To address this, we manually inspected reposts that appeared over 40 times in our dataset, removing content that we have collectively agreed was not harmful. For robustness, we also perform our analyses without considering reposts, as well as only considering unique posts (i.e., reposts are only counted once); the results are qualitatively similar. We also manually inspect 100 random posts before filtering and find 44/100 of these posts are clearly harmful (typically calling someone the n-word, t-slur, or f-slur in a derogatory manner). In contrast, inspecting 100 random posts after filtering, we find 91/100 posts are harmful in the same way. Typically, the misidentified content is either counter-speech (as in "this is a horrible word that should not be used"), reclaimed/reappropriated words (i.e., terms that were considered pejorative in the past, but have since been adopted by those who had been targeted thusly), or pornographic posts (these were often posts by trans sex workers and could also be considered reclaimed words). In total, there were 3K posts in this hate sample.

In addition to measuring the overall volume of hate posts on X, it is important to know whether overall activity on X changed, as an increase in hate speech could simply be due to the fact that users are posting more overall. To accomplish this, we collected a sample representative of English-language X as a whole. Following Hickey et al., we use a set of three common English words to obtain a baseline sample: "thing," "any," and "tell", which are some of the most common words in the English language [93]. Prior to December 1st, 2022, posts containing these words were collected at five randomly sampled five-minute intervals per day. As, due to changes in X's policies, our access to the X API was cut off in the process of conducting this research, our sample of posts in the following months is smaller—the month of December 2022 only contains posts containing the word "thing" (sampled at five random five-minute intervals per day), while posts from 2023 were sampled at one five-minute interval per day. Given that the data for December 2022 is incomplete, we do not consider it when analyzing the baseline sample. Additionally, as the sampling rate of data is 5 minutes per day instead of 25 minutes per day, we subsample all analyses in the 2022 data by 80%. In total, there were 17M posts in this baseline sample. When data before December 1, 2022 were subsampled, this became approximately 4.7M posts with 3.1M accounts.

For both datasets collected, we comply with the X API terms of service for data collection and analysis. The data were publicly collected in the United States and did not require the consent of X users. We remove all personally identifying information from posts prior to analysis, and we only publicly share the ID and timestamp of each post.

Using these samples, we track the total volume per week of posts containing hate speech, as well as the total volume of general posts, and compare the average weekly values before and after Musk's purchase of X. Additionally, to compare engagement with hate posts, we measure

the total number of likes and reposts accumulated by hate posts per week, then measure the average values before and after the purchase. As the documentation on how engagement metrics for reposts are represented by the X API is unclear, we exclude reposts from our dataset for the engagement analysis.

Because we only calculate a few minutes of posts each week, we do not display "weekly" posts, as these would correspond to just a few minutes of data collection. In plots, we estimate the total number of posts each week as the minutes in a week times the number of posts we collected divided by the total minutes we collected these data. We use the same method to upscale engagement counts. This upscaling has no effect on our statistical analysis, and instead makes the plots more interpretable.

## Separating dimensions of hate

To understand the more specific trends in different categories of hate, we first surveyed our dataset of collected posts containing hate speech, observing which terms were the most frequent. We find that three terms appear in the vast majority of posts containing hate speech: the homophobic "f-slur," the racist "n-slur," and the transphobic "t-slur." Example posts containing each slur are shown in S1 File. While slurs of other categories, such as misogynist or antisemitic terms, exist in our dataset, they are not numerous enough to provide a sufficient sample size for assessing differences in their use before and after Musk's purchase. Therefore, we compare the average weekly volume of posts containing the f-slur, n-slur, and t-slur before and after Musk's purchase, labeling these as homophobic, racist, and transphobic posts, respectively. For each category of hate, we observed episodic large spikes in prevalence that were likely due to external events (e.g., transphobic posts surged in early April when Bud Light beer released an advertisement featuring American transgender social media personality Dylan Mulvaney, which led to a substantial boycott of the product). As these punctuated events could skew our results, for each category, we exclude the week with the highest number of posts from that category from consideration. Results are robust whether or not we perform this step.

## Identifying coordinated accounts

We identify coordinated accounts within the baseline sample of posts using three different metrics: hashtag co-occurrence [32], co-reposting similarity [32, 33], and activity similarity [32, 33]. The latter two of these metrics involve making all-to-all pairwise comparisons, therefore larger datasets would be intractable. The first metric measures whether original posts or replies share posts that are very similar by linking two accounts that share at least one post with at least three hashtags in the exact same order. This simple (even simplistic) metric has a high precision in our spot checks, and captures accounts sharing virtually identical content. Co-repost similarity is defined as accounts sharing many identical reposts while controlling for overall repost popularity. For any account that posts at least ten reposts, we record the ID of all reposts then create a TF-IDF vector by calculating each repost ID's "term frequency" (how often it appears for each account) multiplied by their inverse document frequency (how many accounts share this repost). Unlike, e.g., simple repost frequency, TF-IDF gives greater weight to reposts that appear unusually often among a few accounts and less weight to reposts that are popular and broadly shared across X. Accounts whose vectors have a cosine similarity in the top 0.1 percentile are considered coordinated. Finally, activity similarity is defined as accounts posting (reposting or writing original posts, replies, or quote posts) nearly simultaneously. Following previous work [32, 33], we take all accounts with at least 10 posts and bin their post times into 30-minute intervals. Like repost similarity, we map these bins into a

TF-IDF vector, and accounts whose vectors have a cosine similarity in the top 0.1 percentile are considered coordinated. This adds more weight to accounts that are unusually co-active after controlling for times of the day when X accounts are broadly active.

While previous analyses used a lower cutoff of the top 0.5 percentile [33], we find that this results in 48% of active accounts (those with 10 or more posts) being considered "coordinated" based on the activity metric. Similarly, 76% of accounts that post 10 or more retweets are considered "coordinated" based on the co-repost metric. We cannot rule out the possibility that this is because coordinated accounts are typically more active. Because we do not have ground truth as to which accounts are or are not coordinated, we qualitatively evaluated the accounts, finding that a cosine similarity threshold of 0.1% shows a greater precision, with more accounts that push an agenda (e.g., hashtags such as "#TigrayGenocide", related to a civil war in Ethiopia occurring at the time). Only 21% of accounts that post 10 or more times are coordinated via the activity metric, and 29% of accounts that repost more than 10 times are coordinated based on the co-repost metric, which is much more reasonable. For robustness, we further increase the threshold up to the top 0.01% (removing all but a handful of very coordinated accounts) and our results are qualitatively the same. Similarly, our results are consistent when we increase the hashtag threshold from 3 hashtags to 7. Overall, the consistency of our results despite potential variations in precision suggests that our results are not strongly affected by issues of coordination precision.

## Results

### Hate speech remained higher following Musk's purchase

Fig 1 shows the weekly volume of posts containing hate speech from January 1st, 2022, to June 9th, 2023, as well as the weekly volume of posts from the baseline sample (in S1 Fig, we show that these results are robust to words defined as hate speech). Observing the plot, there is a clear increase in hate speech occurring directly before Musk's purchase, and this persists during much of the post-purchase era through March 2023. The large spike in hate speech in April 2023 coincides with a controversy that occurred after Bud Light featured Dylan Mulvaney, a transgender woman, in one of their advertisements [94]. Overall, 37% of the hate posts are composed of replies, 36% are original posts, 19% are reposts, and 7% are quotes. In contrast, the control sample is composed of 28% replies, 14% original posts, 55% reposts, and 3% quotes, i.e., our hate speech dataset has more original posts, replies, and quotes, but fewer reposts, than the baseline dataset.

When comparing the weekly rate of hate speech posts, there is a clear increase in the average number of posts containing hate speech following Musk's purchase. The estimated average number of posts containing hate speech per week before Musk's purchase was 2,179, compared to 3,246 after Musk's purchase, a 50% increase (Mann-Whitney U test, p-value < 0.001). Recognizing that the largest spike in hate speech occurs after Musk's purchase, and is driven by an external event with no clear direct comparison in 2022, we also perform the same analysis excluding the week with the highest number of posts containing hate speech, finding a 44% increase after Musk's purchase (Mann-Whitney U test, p-value < 0.001). Results are also consistent when not considering reposts in the dataset, as well as when we substitute the "toxicity" model for the "identity attack" model (all variations show an approximately 50% increase in posts containing hate speech following Musk's takeover). This contrasts with previous work that found an approximately 100% increase immediately after Musk bought X [26]. The discrepancy is most likely because the previous authors included the r-slur, which we removed (see Methods), and because hate speech was higher in the weeks immediately preceding the purchase. The qualitative conclusion, however, remains the same.

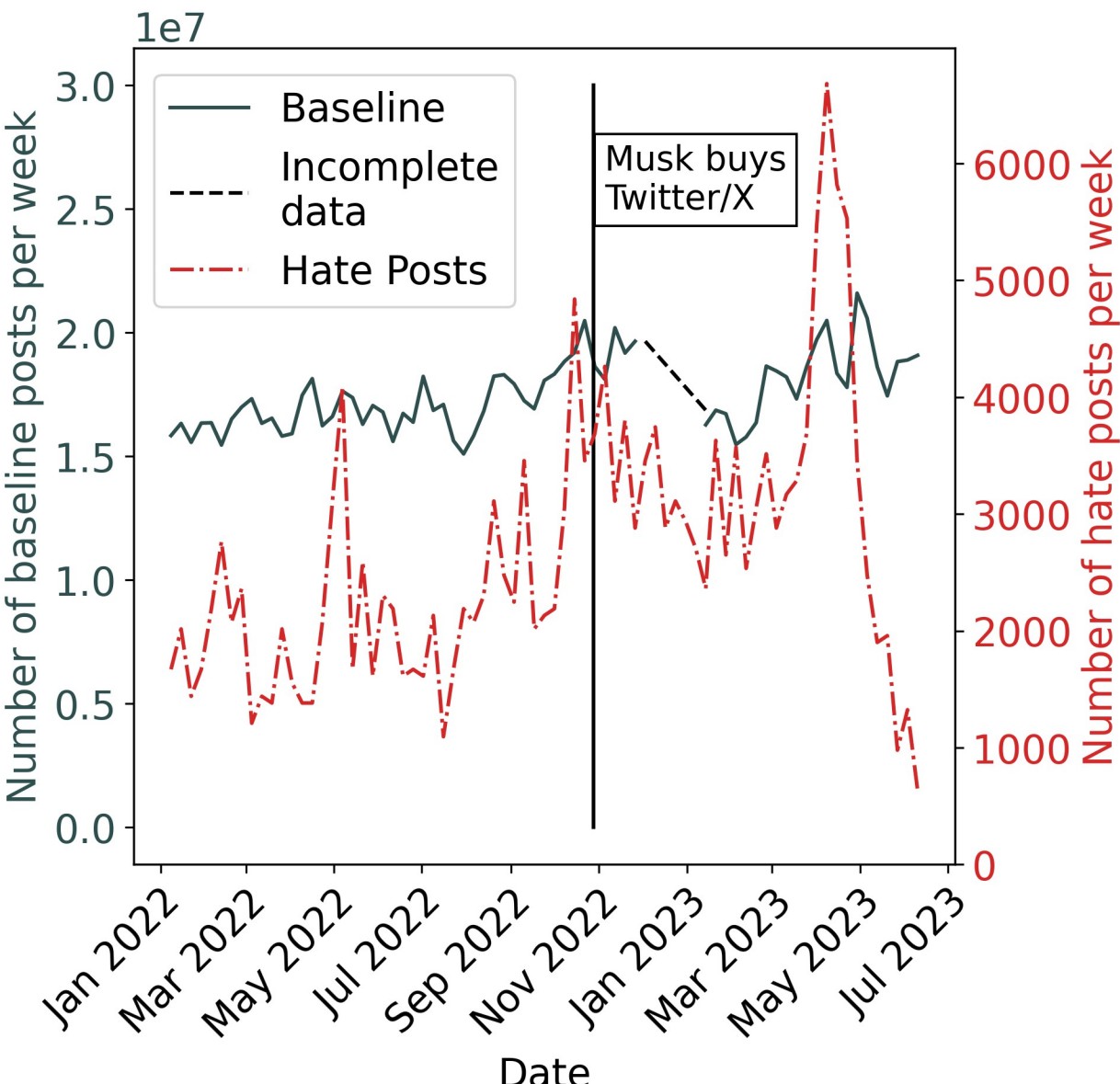

**Fig 1. Hate posts over time.** Y-axis values are estimated values of the total number of posts containing hate speech in a given week.

In addition to posts containing hate speech, the relative number of baseline posts also increases, though to a far lesser degree. The estimated average number of baseline posts per week prior to Musk's purchase is approximately 17.0M, while the estimated average after Musk's purchase is 18.4M. This 8% increase in baseline posts is statistically significant (Mann-Whitney U test, p-value <0.001), but is proportionately vastly smaller than the increase in hate content.

### Engagement with hate posts increases following Musk's purchase

Fig 2 displays the average weekly rate of likes and reposts received by posts containing hate speech before and after Musk's purchase. There is a significant increase in total likes per week,

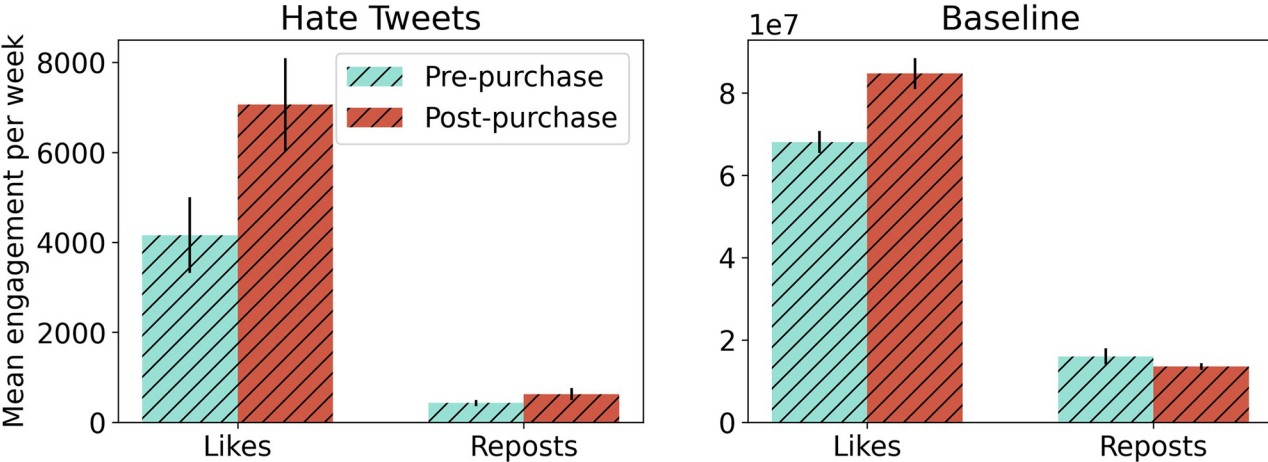

**Fig 2. Mean likes and reposts of (a) hate posts and (b) baseline posts before and after Musk's takeover.** Black vertical lines represent standard errors.

with the average value pre-purchase being 4,158 likes, and the average value post-purchase being 7,066 likes, or 70% higher (Mann-Whitney U test, p-value <0.001). The difference in the average number of reposts, however, is not significant, with 428 reposts per week pre-purchase and 627 reposts per week post-purchase (Mann-Whitney U test, p-value = 0.2).

In addition to measuring the total engagement with posts containing hate speech per week, we also assessed whether the average number of likes and reposts *per post* changed following Musk's purchase. This would address whether the visibility of each hateful post changed, which could suggest that strategies to make posts containing hate speech less visible are working at the level of individual posts. When examining these data, we find that the mean number of likes per post before Musk's purchase was 2.4, while it was 2.7 after Musk's purchase (13% increase; Mann-Whitney U test, p-value = 0.05). The mean reposts per original post did not significantly change (2.45 to 2.38 reposts per post, Mann-Whitney U test, p-value = 0.2).

These results contrast with the baseline dataset, where we only find a 24% increase in likes per week after Musk's purchase (Mann-Whitney U test, p-value < 0.001), as the average estimated number of likes per week before Musk's purchase was 64.0M, while we estimate 84.6M likes per week post-purchase. While this is a statistically significant increase, it is smaller than the increase we observe in the hate dataset. Similarly, after Musk's purchase, we observe an average of 10.0 likes per post, compared to 9.3 likes per post before Musk's purchase, amounting to a 7% increase (Mann-Whitney U test, p-value < 0.001) in the number of likes per post. However, we find a 27% decrease in the number of reposts per post after Musk's purchase (Mann-Whitney U test, p-value < 0.001), with an average of 1.48 reposts per post post-purchase and 2.16 reposts per post pre-purchase. Therefore, baseline posts receive a lower increase in likes per post and a significant decrease in reposts per post, in contrast to the hate dataset. Akin to the hate dataset, however, we find no significant change in the volume of reposts after Musk's purchase compared to before (Mann-Whitney U test, p-value = 0.9).

While we do not have data on the number of views before Musk's purchase, using data following Musk's purchase, we can examine the relationship between our engagement metrics and views in order to determine whether engagement metrics can be used to indirectly track views. Using the views data for messages posted in 2023, we measured the correlation between views and the other engagement metrics. The Spearman correlation between view counts and repost counts is 0.37 (p-value < 0.001), while the Spearman correlation between view counts

and like counts is 0.61 (p-value < 0.001). Therefore, these metrics act as a reasonable proxy of view counts for a given post, with likes being more predictive than reposts.

## Hate speech increases for multiple categories of hate

Fig 3 shows the weekly rates of hate speech before and after Musk's purchase for three dimensions of hate: homophobia, racism, and transphobia. In each case, the weekly rates increase after Musk's purchase. Transphobia exhibits the largest increase, from an average of 115.2 posts containing transphobic slurs per week before Musk's acquisition to an average of 418 after the acquisition, a 260% increase. The average weekly rate of homophobic posts increased from 1,310 per week to 1,737 (a 30% increase), and the average weekly rate of racist posts increased from 579 per week to 822 (a 42% increase). All results are significant (Mann-Whitney U test, p-value < 0.001). We also performed this analysis without considering reposts in the samples, producing qualitatively similar results. Additionally, we performed the analysis using the identity attack model to identify hate speech posts (both with and without including reposts). Across all these variations, results are generally robust. However, when using the identity attack model and not considering reposts, the weekly rate of homophobia does not increase significantly following Musk's acquisition (Mann-Whitney U test, p-value = 0.15). Upon further investigation, it appears that many of the homophobic posts collected consist of the format "*username {f-slur}*," which are classified as toxic by the Perspective API but are not classified as hate speech by the identity attack model (though these posts are borderline; relatedly, note that, viewed psychologically, the use of the f-slur to derogate someone perpetuates homophobia even when the context is one in which sexual orientation is not at issue [95]). We

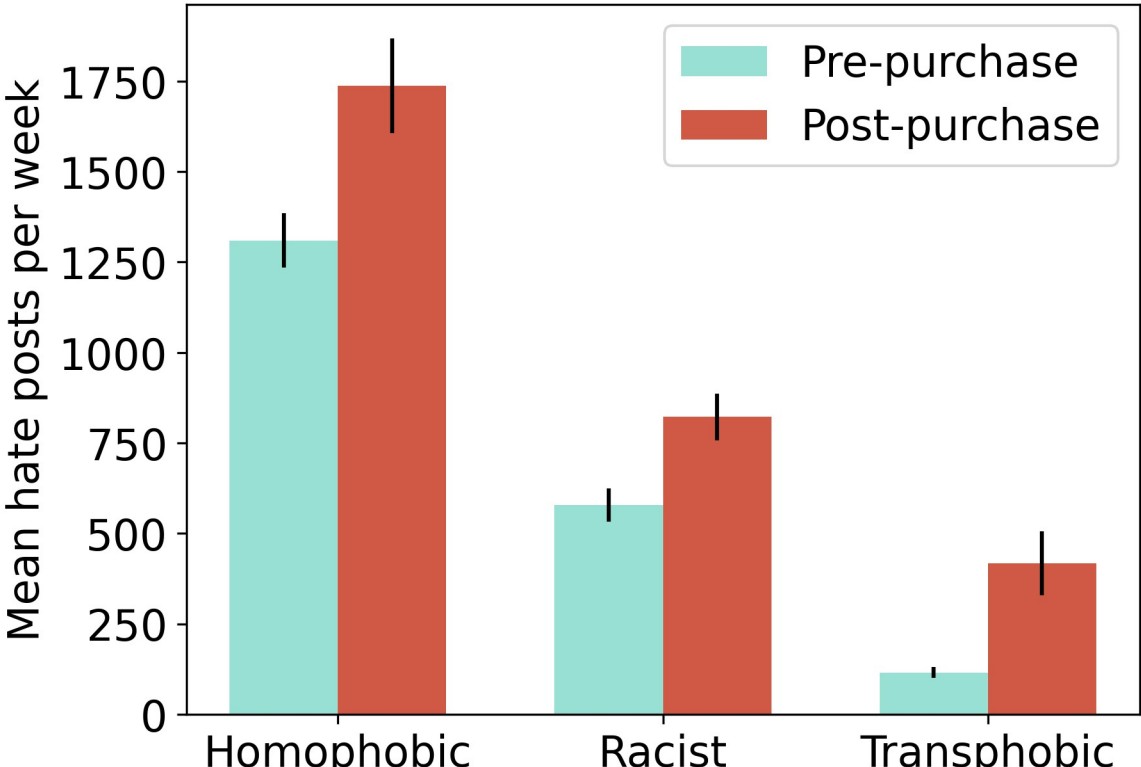

**Fig 3. Increases in hate speech for different dimensions of hate.** Black lines represent standard errors.

therefore conclude that the toxicity model is a better filter for hate speech, and use these results in the main text.

## Activity of inauthentic accounts increased following Musk's purchase

To capture the various ways in which accounts may coordinate their inauthentic activity, we use three different heuristics: hashtag co-occurrence, co-repost similarity, and activity similarity (see Methods section). We first show that these metrics provide qualitatively distinct networks of coordinated accounts, thus their conclusions should not, *a priori*, be similar unless any changes made at X affected coordinated networks more generally. We find that the Jaccard similarities between hashtag co-occurence networks and co-repost or activity networks are at or below 0.005. These networks therefore have very little overlap. In comparison, the Jaccard similarity between co-repost and activity networks is 0.40, thus, while there is some overlap, we still find enough dissimilarity for us to treat the networks as distinct. These metrics each uncover roughly 4K coordinated accounts out of 3.1M accounts in our dataset, but aim for high precision and almost certainly undercount the true number of coordinated accounts. For example, the co-repost and activity metrics only compare coordination among active accounts, constituting roughly 13–20K of all accounts (see details in Methods).

These networks are shown in Fig 4, along with the top hashtags associated with each set of accounts. The hashtag co-occurrence network (Fig 4a) contains 4,846 accounts and the top hashtags of the largest cluster (222 accounts) discusses football: out of 523 posts, #世界杯 (world cup) is mentioned 427 times, #足球 (soccer or football) is mentioned 93 times, and #开云体育 (Kering Sports, a sports forum) is mentioned 78 times. Similarly, the second-largest cluster, with 205 accounts promoted betting: of the 505 posts, #世界杯买球 (World Cup betting) appeared 162 times, #世界杯下注 (World Cup betting) appeared 158 times, and #世界杯外围 (World Cup qualifier) appeared 122 times. Despite capturing tweets containing English words, we see many Chinese hashtags appear. This is because they are embedded in nonsensical English phrases, such as "Measure thing painting admit." in the largest cluster, and "One thing, however, he could do; and he did. He wrote a note #开云 [#Kering] #世界杯 [#World Cup] #AG真人 [AG Real Person]" in the second cluster. The third, meanwhile discusses BTS, a South Korean band (#BTS), and one of their songs, Yet To Come (#YetToCome). While we did not expect a coordinated cluster of accounts discussing BTS or the Football World Cup, we found many of their posts are nearly identical, implying coordination—although we can only speculate on the underlying purpose of these accounts, such as spam to inflate attention directed towards a band or to promote a particular hashtag or betting site.

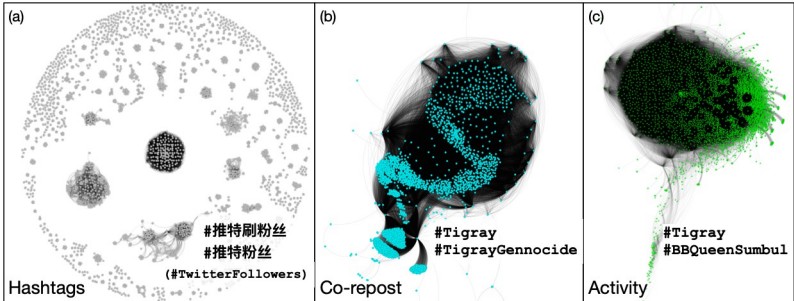

**Fig 4. Coordinated networks' largest components using three different metrics.** (a) hashtag sequence [32], (b) co-repost [32, 33], and (c) activity [32, 33]. In each panel colors are represent nodes (accounts) associated with each metric of coordination. Edges connect nodes if the two accounts are sufficiently similar. The most common non-numerical hashtags are shown to indicate major topics of discussion within each network.

In contrast, the co-repost similarity network (Fig 4b) contains 3,969 accounts; the top hashtags of the largest cluster (3,917 accounts) discuss the Ethiopian civil war (out of 95K posts, #Tigray is mentioned 911 times and #TigrayGenocide, 365 times). Finally, the activity similarity network (Fig 4c) contains 4,374 accounts and its largest cluster (4,125 accounts) promotes similar hashtags, such as #Tigray (658 times) and #TigrayGenocide (275 times) out of 109K posts. Example posts are shown in S1 File. We plot in S2 Fig the Botometer X bot scores for 100 random accounts, 100 hashtag-based coordinated accounts, 100 co-repost-based coordinated accounts, and 100 activity-based coordinated accounts. We find higher bot scores among coordinated accounts compared to random accounts (Mann-Whitney U test, p-values< 0.001), thus confirming the inauthentic behavior of coordinated accounts.

We show the change in coordinated account activity in Fig 5. In Fig 5a, we show the percent change in the number of daily posts in the time period before versus after Musk purchased X. We collect all accounts that posted at least once from April 16 until October 26, inclusive (approximately 6 months before the purchase) and from October 28 until the end of our data collection (June 9, 2023), ignoring December 1–31, 2022, where data are incomplete. We see that there are no significant changes in post activity for coordinated accounts (Mann-Whitney U test, p-values >0.1) and a change of only a few percentage points for non-coordinated accounts (Mann-Whitney U test, p-values = 0.009–0.011 across all metrics). We also find in Fig 5b no significant change in the number of reposts per post (Mann-Whitney U test, p-values>0.08 except for hashtag-based coordination, with a significant increase in reposts per post, p-value< 0.001), and well as in Fig 5c no significant change in the number of likes per posts (Mann-Whitney U test, p-values>0.2), except for hashtag-based coordination (Mann-Whitney U test, p-values = 0.02). Taken together, these results suggest no significant decrease in the attention paid to coordinated accounts. These results are shown to be broadly consistent across all coordinated account definitions and thresholds to distinguish coordinated and non-coordinated accounts, as shown in S3 and S4 Figs.

We separately analyze bot-like behavior as a second metric of inauthentic accounts. In agreement with [26], we see in Fig 6 that there is a significant change in Botometer X bot scores of randomly sampled accounts before and after Musk's purchase, although there is no significant change in bot scores for accounts when the sample is weighted by account activity. Namely, we record the bot scores for 100 random accounts before Musk's purchase, and another 100 after, for any account that posted at least once in our dataset (regardless of account activity). We find significant increases in the bot scores (Mann-Whitney U test p-

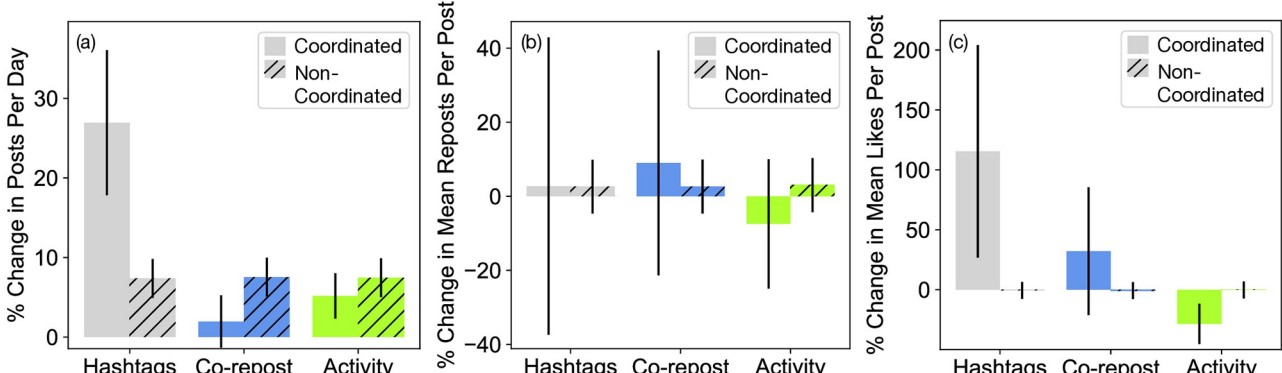

**Fig 5. Changes in coordinated and non-coordinated account activity and interactions after Musk's purchase.** (a) Percent change in posts per day, (b) change in mean reposts per post, and (c) change in mean likes per post before versus after Musk's purchase. Black bars represent standard errors. We use 3 different coordination metrics: hashtag sequence [32], co-repost [32, 33], and activity [32, 33].

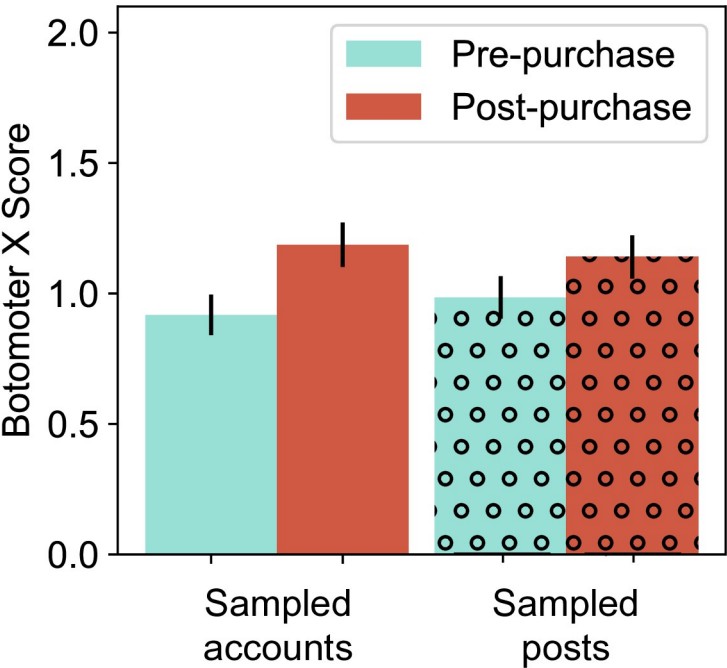

**Fig 6. Bot scores for random accounts before and after Musk's purchase (bot scores vary between 0, organic users, and 5, clear bot activity).**

value = 0.01). We separately collected accounts associated with 100 random posts before and another 100 after the purchase (thus giving more weight to highly active accounts). We do not find a significant increase in bot scores (Mann-Whitney U test p-value = 0.11). These two results imply that the number of active bot accounts may have increased, although the overall number of posts by bot accounts is not necessarily higher than before Musk's purchase.

Finally, we analyzed one more potential inauthentic set of accounts: those that share cashtags, which are tags such as "$ETH" (Etherium) that represent cryptocurrency or stocks. While accounts can share cashtags benignly, and indeed there is no reason to automatically assume an account promoting Etherium is inauthentic, there are nonetheless many inauthentic accounts that use cashtags within spam posts. This is confirmed both by a higher bot score for accounts that share cashtags (an average of 1.55 among 100 sampled accounts associated with cashtags versus 0.98 among 100 random accounts, Mann-Whitney U test, p-value < 0.001), as well as through the far greater use of cashtags among hashtag-based coordinated accounts (see S5 Fig). For example, one cluster of 12 coordinated accounts reposts messages such as "$OGGY @XXX is the next big thing in the crypto world! #OGGYINU #OGGY #BSC". Given near-duplicate messages of this form, we can surmise that these posts are meant to bring attention to a particular cryptocurrency. While this is a weak metric of inauthentic account activity, their use has very clearly spiked, as seen in Fig 7. The slope of the best-fit line is significantly negative before Musk's purchase (regression p-value<0.0001), but significantly positive after the purchase (regression p-value<0.0001).

## Discussion

We analyzed hate speech and inauthentic accounts on X before and after Elon Musk acquired the site, seeking to determine the accuracy of public statements about the platform made by

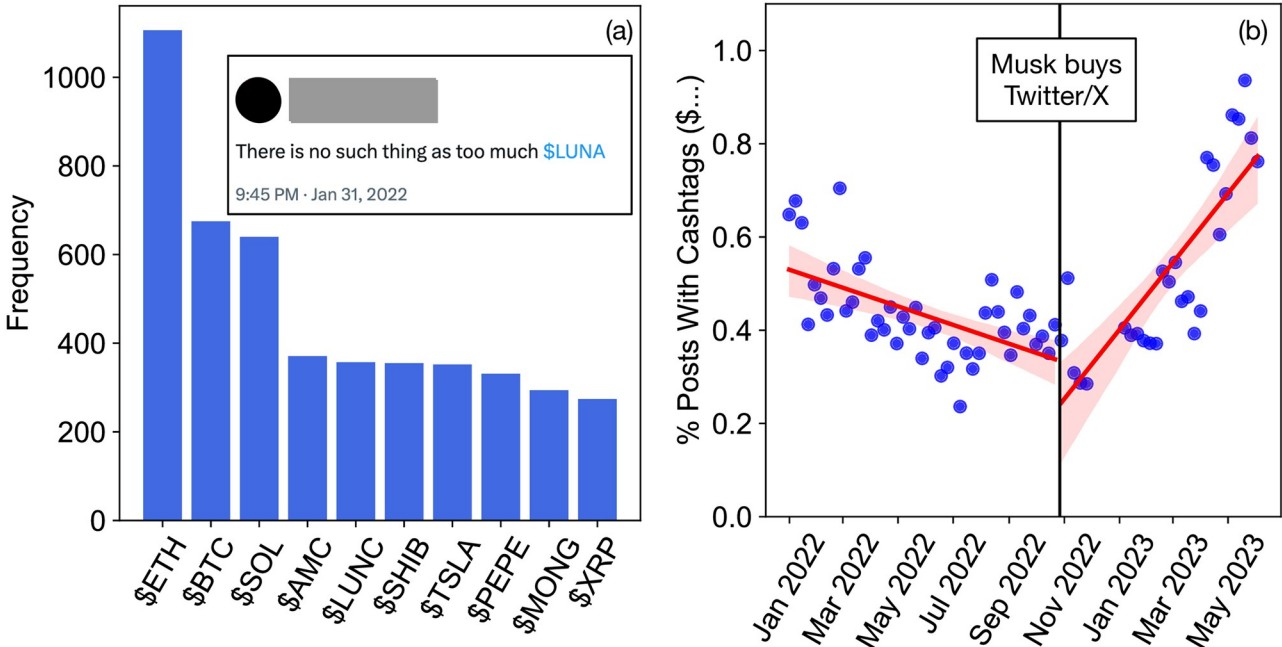

**Fig 7. Cashtags become substantially more popular after Musk purchased X.** (a) Most frequent cashtags (inset is an example post). (b) Decrease in cashtag frequency before Musk purchased X, and substantial increase after Musk's purchase.

Musk and X about how the site has changed. Contrary to a number of such statements, we find significant increases in hate speech that targets various protected groups, as well as increases in the visibility of hate speech. Moreover, we find that, following Musk's acquisition, coordinated account activity associated with information campaigns increased more than non-coordinated activity. Because we are not privy to many facets of the internal policies, procedures, and organizational composition of X following Musk's acquisition, we cannot derive definitive causal claims from our observations. Nevertheless, in the following sections we speculate on factors that may underlie these patterns, and discuss their potential consequences.

### Exogenous events contribute to the increase in hate speech, policies may sustain it

As noted, we are unable to definitively determine what caused the increased rate of hate speech following Musk's purchase of X. While specific actions made by the company directly following the acquisition, such as reducing the size of the trust and safety team [4], offer a tempting explanation, we cannot evaluate the impact of specific events or policies implemented following the acquisition. In fact, as indicated in Fig 1, there was an increase in hate speech just before Musk bought the platform, suggesting that the high levels of hate speech directly after his purchase were not due to any specific change at X itself, but rather may have stemmed from users having anticipated that such behavior would be allowed, as Musk stated he would decrease content moderation upon acquiring the platform. X also reinstated several influential accounts previously suspended for promoting hate [96]. Additionally, many external events outside the control of individual social media platforms may have resulted in more hate speech on a platform—for example, a U.S. midterm election occurred shortly after Musk's acquisition, and there were likely many posts containing hate speech related to that topic. Nevertheless,

these potential explanations certainly do not rule out the possibility that direct changes to the platform made by the organization had an impact. For example, in April 2023 protections for transgender individuals were deleted from the hateful conduct policy, a change that plausibly led to fewer transphobic posts being removed from the platform [97].

When qualitatively observing the posts containing hate speech, we noticed that posts in some categories exhibit themes specific to that category, themes that go beyond the denigration of the targeted identities. For example, while both transphobic and homophobic posts often repeated the moldy trope that people who are transgender or gay suffer from mental illness, relative to homophobic posts, transphobic posts more often expressed the poster's political views. Many users who posted transphobic material stated their support for, or antipathy toward, a given politician based on that politician's position on transgender rights. In contrast, while racist posts in our sample often asserted similarly longstanding hateful stereotypes, unlike transphobic posts, racist posts did not frequently connect such statements to evaluations of politicians. Hence, while a substantial portion of hate speech on X consists of reiterations of perennial derogatory stereotypes about the target group or groups, there is variation across categories of hate speech as to the extent to which hate speech is associated with partisan political stances. The latter presumably reflects the political landscape at the time that posts are written, including the prominence of particular types of discrimination in contemporaneous "culture-war" debates in the public sphere.

## Hate posts are more visible despite the algorithm

Since Musk's acquisition of the platform, X has published their stance on hate speech, claiming that the visibility of posts containing hate speech will drop even if more of such content exists on X [21]. This accompanies claims made by the company that engagement on posts containing hate speech has dropped since the acquisition [22, 23]. Although we do not have data on post view counts prior to Musk's acquisition, our results examining a proxy of visibility, likes (Fig 2), suggest that the overall visibility of hate speech *increased* following Musk's acquisition. X's algorithm aims to show content that people will like and which is similar to their interests [98], thus our results suggest that either hate speech was insufficiently downranked as its frequency increased; previously passive hate users became more active (independent of the algorithm); the algorithm is unintentionally promoting hate speech to users who like such content; or some combination of these possibilities.

## No reduction in inauthentic account activity

We find no significant change in coordinated account activity or engagement. That said, we see an increase in the number of active bot-like accounts, and a substantial upward trend in cryptocurrency posts, which often contain spam. All of these results suggest no decrease, and a potential increase, in activity among inauthentic accounts. This could have grave consequences not merely for user enjoyment of the platform, but also for potential scam victims and the stability of democracies in the face of disinformation. Moreover, the rise of LLMs makes inauthentic speech even easier to produce, thus we have no reason to believe that inauthentic activity has decreased in the period since our data were collected. More work is needed to understand the impact of inauthentic accounts in the lead-up to important U.S. national elections in 2024.

## Limitations and future directions

While our study documents trends in coordinated activity and hate speech on X over time, significant questions remain regarding the impacts of the trends that we observe; relatedly, it is

important to note that our study is subject to a number of limitations. Here, we elaborate on those limitations and offer suggestions for future research.

## Policies on X

As we have noted, we have little information concerning when and how X enforced various moderation strategies. For example, it is possible that specific moderation strategies aiming to reduce hate took considerable time to implement, and thus were not in effect during the majority of our observed period. However, we do check claims made publicly by X about the production and visibility of hate speech following Musk's acquisition, finding no evidence to support the claim that either decreased. Our work demonstrates the importance of independently evaluating claims made by social media companies about the content on their platforms, as such statements are often accompanied neither by evidence nor by transparent information regarding how such conclusions were reached.

## Musk's impact on hate speech in different languages

While we provide a thorough analysis of how English-language hate speech changed in the context of Musk's purchase, many other languages are used on X—indeed, only 31% of posts on the platform are in English, with languages such as Japanese and Spanish also being prominent [99]. Although language and culture are not isomorphic, given that, as reflected in such linguistic diversity, individuals from many cultures use X, it is noteworthy that culture may influence users' responses to Musk's purchase of the platform. Accordingly, in the future, it will be important to expand analyses such as ours to include greater linguistic and cultural diversity.

## Prevalence of covert hate speech on X

The hate speech measured in this study is generally *overt*, as its intended meaning is clear to the general population on X. However, as social media platforms have historically limited this type of speech, many extremist groups use coded language to spread hate, helping them to maintain plausible deniability [100, 101]. Future work should determine how the relative concentration of covert hate speech on X changed following Musk's purchase, and how it relates to the presence of overt hate speech on the platform. We note in passing that this is a challenging task, as, in order to stay one step ahead of moderators, users of covert hate speech are constantly both coining neologisms and coordinating on implicit meanings for seemingly innocuous terms.

## Better coordination detection

While we have used three different state-of-the-art metrics to detect coordination, each of these methods is subject to limitations. First, the metrics can misclassify accounts. For example. the activity metric misidentifies @NBA, the X account of the National Basketball Association, as part of the coordinated network. This example showcases how these metrics can make glaring mistakes. That being said, because these metrics aim to have high precision, many coordinated accounts can be misidentified as authentic, meaning that we do not have a full picture of coordinated account activity. It is therefore crucial to improve coordination detection and reevaluate our findings in the future, employing new unsupervised and supervised methods [71].

## A pressing need

Today, the world faces an unprecedented combination of large-scale emergencies and global interconnectivity. Transmissible disease arising in one location spreads to every corner of the planet. Violent conflict is both fueled by and disrupts economies half a world away. And climate change affects us all. Challenges such as these can only be met through cooperation, coordination, and compromise. Although social media platforms have the potential to further prosocial orientations and enhance support for constructive collective actions, they can also serve as vehicles for the dissemination of hate and disinformation, pitting one group against another, splintering communities, polities, and whole nations in ways that both lead directly to suffering and preclude solutions to global challenges. Moreover, the information landscape has become so large, complex, and rapidly changing that it is impossible for objective policymakers or journalists to monitor it unaided. We believe that researchers have an ethical obligation to develop tools that can illuminate this landscape. Even as we recognize that our own efforts are subject to limitations, we hope that they nonetheless contribute to this task, and inspire others to improve upon our work.

## Supporting information

**S1 File. Example posts and robustness checks.** We show example hate speech and coordinated account posts, as well as robustness checks of our results.
(PDF)

**S1 Fig. Hate posts over time when including posts that contain the r-slur.**
(PNG)

**S2 Fig. Bot scores for coordinated and random accounts (bot scores vary between 0, organic users, and 5, clear bot activity).** Bot scores are significantly higher in coordinated accounts (above brackets are Mann-Whitney U test p-values) when we measure 100 random accounts, 100 accounts coordinated by hashtag sequences, 100 accounts coordinated by co-repost behavior, and 100 accounts coordinated by activity. All accounts are sampled at random regardless of their activity.
(PDF)

**S3 Fig. Changes in coordinated and non-coordinated account activity and interactions after Musk's purchase.** We use 3 different coordination metrics: hashtag sequence [32], co-repost [32, 33], and activity [32, 33]. Accounts are coordinated if they share a sequence of at least 5 hashtags in a post or have a co-retweet or activity cosine similarity in the top 0.5 percentile. (a) Percent change in posts per day, (b) change in mean reposts per post, and (c) change in mean likes per post before versus after Musk purchased X. Changes are for not significant for coordinated accounts in (a) (Mann-Whitney U test p-values > 0.2) except for hashtag-based coordinated accounts (p-value = 0.04), but are significant for non-coordinated accounts (Mann-Whitney U test p-values <0.008). No changes are statistically significant for (b) or (c) (Mann-Whitney U test p-values ≥0.2) except for the change in reposts per post for hashtag-based coordinated accounts (Mann-Whitney U test p-value = 0.009). Error bars are standard errors. Because of the differences between how standard error and Mann-Whitney U test are calculated, there can be standard errors that overlap with zero and the results could still be statistically significant.
(PDF)

**S4 Fig. Changes in coordinated and non-coordinated account activity and interactions after Musk's purchase.** We use 3 different coordination metrics: hashtag sequence [32], co-

repost [32, 33], and activity [32, 33]. Accounts are coordinated if they share a sequence of at least 7 hashtags in a post or have a co-retweet or activity cosine similarity in the top 0.01 percentile. (a) Percent change in posts per day, (b) change in mean reposts per post, and (c) change in mean likes per post before versus after Musk purchased X. Changes are for significant for all data (Mann-Whitney U test p-values <0.05) except for co-repost coordinated accounts (Mann-Whitney U test p-values = 0.38). No changes are statistically significant for (b) or (c) (Mann-Whitney U test p-values > 0.1) except for the change in reposts per post for hashtag based coordinated accounts (Mann-Whitney U test p-value = 0.002). Error bars are standard errors. Because of the differences between how standard error and Mann-Whitney U test are calculated, there can be standard errors that overlap with zero and the results could still be statistically significant.
(PDF)

**S5 Fig. Percent of posts that contain cashtags for coordinated and non-coordinated accounts.** (a) 3-hashtag cutoff and cosine similarity percentage greater than 99.9%, (b) 5-hashtag cutoff and cosine similarity percentage greater than 99.5%, (c) 7-hashtag cutoff and cosine similarity percentage greater than 99.99%. We observe cashtags are significantly more common among hashtag similarity-based coordinated accounts.
(PDF)

## Author Contributions

**Conceptualization:** Daniel M. T. Fessler, Keith Burghardt.

**Data curation:** Daniel Hickey, Keith Burghardt.

**Formal analysis:** Daniel Hickey.

**Funding acquisition:** Kristina Lerman, Keith Burghardt.

**Investigation:** Daniel Hickey, Keith Burghardt.

**Methodology:** Daniel Hickey, Keith Burghardt.

**Project administration:** Kristina Lerman.

**Resources:** Keith Burghardt.

**Software:** Daniel Hickey.

**Supervision:** Daniel M. T. Fessler.

**Validation:** Keith Burghardt.

**Visualization:** Daniel Hickey, Keith Burghardt.

**Writing – original draft:** Daniel Hickey, Daniel M. T. Fessler, Kristina Lerman, Keith Burghardt.

**Writing – review & editing:** Daniel Hickey, Daniel M. T. Fessler, Kristina Lerman, Keith Burghardt.

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
