## [Decision Letter · Decision Letter 0]

23 Aug 2024

PONE-D-24-20764X Under Musk’s Leadership: More Hate and No Reduction in Inauthentic ActivityPLOS ONE

Dear Dr. Burghardt,

Thank you for submitting your manuscript to PLOS ONE. After careful consideration, we feel that it has merit but does not fully meet PLOS ONE’s publication criteria as it currently stands. Therefore, we invite you to submit a slightly revised version of the manuscript that addresses the points raised during the review process. Specifically, the reviewers suggested some improvements concerning the research questions addressed in the manuscript and asked for a very limited set of additional results that could further strengthen the paper. Moreover, they requested a few clarifications about the adopted methodology and part of the results. Overall, the required edits should be pretty straightforward to complete and we look forward to receiving the revised version of this interesting manuscript. Comments from PLOS Editorial Office: We note that one or more reviewers has recommended that you cite specific previously published works. As always, we recommend that you please review and evaluate the requested works to determine whether they are relevant and should be cited. It is not a requirement to cite these works. We appreciate your attention to this request.

We look forward to receiving your revised manuscript.

Kind regards,

Stefano Cresci

Academic Editor

PLOS ONE

Journal Requirements:

2. In your Methods section, please include additional information about your dataset and ensure that you have included a statement specifying whether the collection and analysis method complied with the terms and conditions for the source of the data.

"Funding for this work is provided through NSF (award #2051101), and through DARPA (awards #HR0011260595 and #HR001121C0169)."

"DH is funded through the National Science Foundation (award #2051101; https://www.nsf.gov/), who did not play any role in the study design, data collection and analysis, decision to publish, or preparation of the manuscript.

KL and KB are funded through the Defense Advanced Research Projects Agency (awards #HR0011260595 and #HR001121C0169; https://www.darpa.mil/), who did not play any role in the study design, data collection and analysis, decision to publish, or preparation of the manuscript."

Reviewers' comments:

Reviewer's Responses to Questions

**Comments to the Author**

1. Is the manuscript technically sound, and do the data support the conclusions?

Reviewer #1: Yes

Reviewer #2: Yes

2. Has the statistical analysis been performed appropriately and rigorously? 

Reviewer #1: Yes

Reviewer #2: Yes

3. Have the authors made all data underlying the findings in their manuscript fully available?

Reviewer #1: Yes

Reviewer #2: Yes

4. Is the manuscript presented in an intelligible fashion and written in standard English?

Reviewer #1: Yes

Reviewer #2: Yes

5. Review Comments to the Author

Reviewer #1: Authors examine material posted on Twitter/X from the beginning of 2022 through June 2023, the period that includes Musk’s full tenure as CEO. Through an analysis of hate speech level, coordinated behavior detection and inauthenticity of the accounts, they investigate if Musk’s purchase of X is correlated with hate speech, and inauthentic account activity. The experiments are performed according to a scientific procedure and the paper is correctly english-written. However, some suggestions:

As you mention in the limitation, you are unable to definitively determine what caused the increased rate of hate speech following Musk’s purchase of X. Given this observation, I would make some comments on the results less assertive. I would start with the title, which states “More Hate”. As far as I understood, there has been a peak in hate speech even previously Musk’s purchase. And there is only a very high peak after the purchase and no clear long trend.

In account coordination, you state “Previous work found relatively little correlation between bots and coordination [33]”. Actually Nizzoli et al. [33] is a particular case of study, while the idea of coordination on social media originated from botnets evolution [1], until Facebook proposed the concept of Coordinated Inauthentic Behavior (which includes the concept of inauthenticity and coordination). As you stated, an example is the interplay between coordination and information operation [2], which could make use of bots.

In Figure 2, you show the mean engagement hate speech per week. I would ask you why you do not show the same plot for the baseline. Due to possible offline, external events it can happen a general increase in the engagement both for hate speech posts and baseline

[1] Mannocci, et al. "Mulbot: Unsupervised bot detection based on multivariate time series." 2022 IEEE international conference on big data (Big Data). IEEE, 2022.

[2] Cima, et al. "Coordinated Behavior in Information Operations on Twitter." IEEE Access (2024).

Reviewer #2: Thank you for allowing me to read this work on the prevalence of hate speech, relative engagement (including exposure to such content), and coordinated and inauthentic behavior on Twitter before and after Elon Musk's takeover. The topic is undoubtedly relevant, and the analysis is methodologically sound and interesting in its findings. Overall, I believe the paper is nearly ready for publication, and I commend the authors for producing a manuscript of such high quality. I also appreciated the methodological choices and the additional analyses conducted to test the robustness of various results.

I have just a few suggestions.

I recommend integrating the research questions (for example, the first one) with a reference to the question on engagement (likes), which is currently missing, even though it is an area of investigation that is extensively addressed and revisited in the discussion. Alternatively, a new research question could be added to cover this aspect. Additionally, this research question needs to be introduced adequately. At present, immediately after the first research question, there is a section dedicated to discussing topics such as engagement, likes, and exposure (views). These are different concepts, and their relationships should be addressed more clearly, as they currently lack depth. The authors seem primarily interested in user exposure to problematic content to test the assertion regarding freedom of speech but not reach. However, they mainly use likes as a proxy for exposure, as reported in the results where the correlation between the two measures is described. When introducing the research question, it should be clarified whether the focus is on engagement, exposure, or both, or if likes are used as a proxy for exposure. In that case, the choice should be justified, and the potential and limitations of this measure should be described. These aspects should be outlined when introducing the research question and in the method section, wherever appropriate, depending on the aspect being discussed.

When presenting the results, many hashtags appear in Eastern characters and languages, despite the authors analyzing English-language content. It would be helpful to elaborate on this aspect in more detail.

Clarifications and minor points:

As someone interested in gender issues and political communication on social media, I found the data on hate speech against transgender and homosexual individuals particularly compelling. I believe this finding could make the paper appealing to researchers working in this field, as these measures help to better understand the social and cultural context and the characteristic communication of a platform. Perhaps the authors could add some further qualitative details on the type of content that constitutes a significant portion of the hate speech on platform X, who share it, and any other details that might be relevant.

Could you provide more details on the methodology used to identify coordinated behavior? For example, for readers who may not be entirely familiar with these methods, it would be helpful to specify how the total volume of reposts is controlled when identifying coordinated actors. Additionally, explaining what TF-IDF represents and why this particular representation was chosen over others would be beneficial.

In the methods section, it is stated that sexual content is filtered out, but it is also mentioned that content with a sexual score above 0.3 is retained. This should be the other way around. Could you please verify?

6. PLOS authors have the option to publish the peer review history of their article (what does this mean?). If published, this will include your full peer review and any attached files.

Reviewer #1: No

Reviewer #2: No

---

## [Author Response · Author response to Decision Letter 0]

15 Oct 2024

Editor

Editor critique

"Specifically, the reviewers suggested some improvements concerning the research questions addressed in the manuscript and asked for a very limited set of additional results that could further strengthen the paper." 

We thank the reviewers for these suggestions and have substantially re-written the manuscript and updated our results to address these concerns.

"Moreover, they requested a few clarifications about the adopted methodology and part of the results."

We have added additional text to improve the clarity of the methodology and results sections.

Reviewer #1

Reviewer critique

"As you mention in the limitation, you are unable to definitively determine what caused the increased rate of hate speech following Musk’s purchase of X. Given this observation, I would make some comments on the results less assertive. I would start with the title, which states “More Hate”. As far as I understood, there has been a peak in hate speech even previously Musk’s purchase. And there is only a very high peak after the purchase and no clear long trend."

We have toned down our claims and the description of our results throughout the manuscript, and have changed the title to “Substantial Hate” (rather than the previous “More Hate”) to address this concern.

"In account coordination, you state “Previous work found relatively little correlation between bots and coordination [33]”. Actually Nizzoli et al. [33] is a particular case of study, while the idea of coordination on social media originated from botnets evolution [1], until Facebook proposed the concept of Coordinated Inauthentic Behavior (which includes the concept of inauthenticity and coordination). As you stated, an example is the interplay between coordination and information operation [2], which could make use of bots.

[1] Mannocci, et al. "Mulbot: Unsupervised bot detection based on multivariate time series." 2022 IEEE international conference on big data (Big Data). IEEE, 2022.

[2] Cima, et al. "Coordinated Behavior in Information Operations on Twitter." IEEE Access (2024).

We thank the reviewer for this critique and have added these references to the manuscript. We have also added new references to contextualize hate speech and misinformation (Saha et al., 2023; Martel and Rand, 2023; Schneider et al., 2023; Kennedy et al., 2022)."

We have also clarified the relevant section of the paper, which now reads as follows:

“There is often a strong overlap between the presence of bots and patterns of coordination (although a study of coordinated activity during the 2019 UK general election found relatively little correlation between bots and coordination).”

Added references:

Kennedy, B., Jin, X., Davani, A. M., Dehghani, M., & Ren, X. (2020). Contextualizing hate speech classifiers with post-hoc explanation. arXiv preprint arXiv:2005.02439.

Martel, C., & Rand, D. G. (2023). Misinformation warning labels are widely effective: A review of warning effects and their moderating features. Current Opinion in Psychology, 101710.

Saha, P., Garimella, K., Kalyan, N. K., Pandey, S. K., Meher, P. M., Mathew, B., & Mukherjee, A. (2023). On the rise of fear speech in online social media. Proceedings of the National Academy of Sciences, 120(11), e2212270120.

Schneider, P. J., & Rizoiu, M. A. (2023). The effectiveness of moderating harmful online content. Proceedings of the National Academy of Sciences, 120(34), e2307360120.

"In Figure 2, you show the mean engagement hate speech per week. I would ask you why you do not show the same plot for the baseline. Due to possible offline, external events it can happen a general increase in the engagement both for hate speech posts and baseline"

This is an excellent point; we have updated Figure 2 to include the same plot for the baseline.

Reviewer #2

"Thank you for allowing me to read this work on the prevalence of hate speech, relative engagement (including exposure to such content), and coordinated and inauthentic behavior on Twitter before and after Elon Musk's takeover. The topic is undoubtedly relevant, and the analysis is methodologically sound and interesting in its findings. Overall, I believe the paper is nearly ready for publication, and I commend the authors for producing a manuscript of such high quality. I also appreciated the methodological choices and the additional analyses conducted to test the robustness of various results."

We thank the reviewer for the positive feedback. 

"I recommend integrating the research questions (for example, the first one) with a reference to the question on engagement (likes), which is currently missing, even though it is an area of investigation that is extensively addressed and revisited in the discussion. Alternatively, a new research question could be added to cover this aspect."

We have created new research questions to address this apt suggestion, as follows:

How did Musk's purchase of X, and subsequent policy changes, correlate with the volume of hate speech on X?

How did the level of engagement with posts containing hate speech change on X following Musk's purchase of the platform?

How did Musk's purchase of X, and subsequent policy changes, correlate with inauthentic account activity on X?

How did engagement with posts made by inauthentic accounts change following Musk's purchase of X?

"Additionally, this research question needs to be introduced adequately. At present, immediately after the first research question, there is a section dedicated to discussing topics such as engagement, likes, and exposure (views). These are different concepts, and their relationships should be addressed more clearly, as they currently lack depth. The authors seem primarily interested in user exposure to problematic content to test the assertion regarding freedom of speech but not reach. However, they mainly use likes as a proxy for exposure, as reported in the results where the correlation between the two measures is described. When introducing the research question, it should be clarified whether the focus is on engagement, exposure, or both, or if likes are used as a proxy for exposure. In that case, the choice should be justified, and the potential and limitations of this measure should be described. These aspects should be outlined when introducing the research question and in the method section, wherever appropriate, depending on the aspect being discussed."

We have now clarified our discussion of these topics in the Introduction as follows:

“We use the term ‘engagement’ to refer to the number of likes and reposts received by a given post on X. We are primarily interested in users' exposure to hate speech on the platform, yet we do not have data on the number of views each post received before Musk's purchase. Engagement metrics can therefore act as a reasonable proxy for the visibility of posts on X. While we cannot be certain that the number of likes received by a certain post directly corresponds to the number of views received by that post, we nevertheless find that these two metrics are strongly correlated. Previous independent reports show that hate speech has increased since Musk purchased X; however, this prior research did not explore changes in the prevalence of hate speech over the longer term, and did not explore whether an increase in hate posts corresponds to an increase in engagement with hate.” 

"When presenting the results, many hashtags appear in Eastern characters and languages, despite the authors analyzing English-language content. It would be helpful to elaborate on this aspect in more detail."

We clarify in the Results section that, “Despite capturing tweets containing English words, we see many Chinese hashtags appear. This is because these hashtags are embedded in nonsensical English phrases such as ``Measure thing painting admit.'' in the largest cluster, and ``One thing, however, he could do; and he did. He wrote a note #开云 [#Kering] #世界杯 [ #World Cup] #AG真人 [AG Real Person]'' in the second cluster.”

"As someone interested in gender issues and political communication on social media, I found the data on hate speech against transgender and homosexual individuals particularly compelling. I believe this finding could make the paper appealing to researchers working in this field, as these measures help to better understand the social and cultural context and the characteristic communication of a platform. Perhaps the authors could add some further qualitative details on the type of content that constitutes a significant portion of the hate speech on platform X, who share it, and any other details that might be relevant."

We appreciate the reviewer’s point, and have therefore added substantial additional text to the Discussion section, as follows:

“When qualitatively observing the posts containing hate speech, we noticed that posts in some categories exhibit themes specific to that category, themes that go beyond the denigration of the targeted identities. For example, while both transphobic and homophobic posts often repeated the moldy trope that people who are transgender or gay suffer from mental illness, relative to homophobic posts, transphobic posts more often expressed the poster's political views. Many users who posted transphobic material stated their support for, or antipathy toward, a given politician based on that politician's position on transgender rights. In contrast, while racist posts in our sample often asserted similarly longstanding hateful stereotypes, unlike transphobic posts, racist posts did not frequently connect such statements to evaluations of politicians. Hence, while a substantial portion of hate speech on X consists of reiterations of perennial derogatory stereotypes about the target group or groups, there is variation across categories of hate speech as to the extent to which hate speech is associated with partisan political stances. The latter presumably reflects the political landscape at the time that posts are written, including the prominence of particular types of discrimination in contemporaneous ‘culture-war’ debates in the public sphere.”

"Could you provide more details on the methodology used to identify coordinated behavior? For example, for readers who may not be entirely familiar with these methods, it would be helpful to specify how the total volume of reposts is controlled when identifying coordinated actors. Additionally, explaining what TF-IDF represents and why this particular representation was chosen over others would be beneficial."

We have now added more detail in the Methods section to describe the coordination detection methodology, which we quote below:

“Co-repost similarity is defined as accounts sharing many identical reposts while controlling for overall repost popularity. For any account that posts at least ten reposts, we record the ID of all reposts then create a TF-IDF vector by calculating each repost ID's ‘term frequency’ (how often it appears for each account) multiplied by their inverse document frequency (how many accounts share this repost). Unlike, e.g., simple repost frequency, TF-IDF gives greater weight to reposts that appear unusually often among a few accounts and less weight to reposts that are popular and broadly shared across X. Accounts whose vectors have a cosine similarity in the top 0.1 percentile are considered coordinated. Finally, activity similarity is defined as accounts posting (reposting or writing original posts, replies, or quote posts) nearly simultaneously. Following previous work, we take all accounts with at least 10 posts and bin their post times into 30-minute intervals. Like repost similarity, we map these bins into a TF-IDF vector, and accounts whose vectors have a cosine similarity in the top 0.1 percentile are considered coordinated. This adds more weight to accounts that are unusually co-active after controlling for times of the day when X accounts are broadly active.”

"In the methods section, it is stated that sexual content is filtered out, but it is also mentioned that content with a sexual score above 0.3 is retained. This should be the other way around. Could you please verify?"

Thank you for pointing this out; we have corrected the text as follows: “We then use the Perspective API to remove posts with toxicity confidence below 0.7 and with sexually explicit confidence above 0.3.”

---

## [Decision Letter · Decision Letter 1]

22 Oct 2024

X Under Musk’s Leadership: Substantial Hate and No Reduction in Inauthentic Activity

PONE-D-24-20764R1

Dear Dr. Burghardt,

We’re pleased to inform you that your manuscript has been judged scientifically suitable for publication and will be formally accepted for publication once it meets all outstanding technical requirements.

Kind regards,

Stefano Cresci

Academic Editor

PLOS ONE

Additional Editor Comments (optional):

Thank you for your work on the revised manuscript. Both reviewers agree that you have addressed all of their comments.

Therefore the paper can now be accepted for publication, congratulations!

Reviewers' comments:

Reviewer's Responses to Questions

**Comments to the Author**

1. If the authors have adequately addressed your comments raised in a previous round of review and you feel that this manuscript is now acceptable for publication, you may indicate that here to bypass the “Comments to the Author” section, enter your conflict of interest statement in the “Confidential to Editor” section, and submit your "Accept" recommendation.

Reviewer #1: All comments have been addressed

Reviewer #2: All comments have been addressed

2. Is the manuscript technically sound, and do the data support the conclusions?

Reviewer #1: Yes

Reviewer #2: Yes

3. Has the statistical analysis been performed appropriately and rigorously? 

Reviewer #1: N/A

Reviewer #2: Yes

4. Have the authors made all data underlying the findings in their manuscript fully available?

Reviewer #1: Yes

Reviewer #2: Yes

5. Is the manuscript presented in an intelligible fashion and written in standard English?

Reviewer #1: Yes

Reviewer #2: Yes

6. Review Comments to the Author

Reviewer #1: The suggestions have been addressed, but all the Figure, except Figure 2, which is the one modified, are disappeared. Please, add all the figures in the final version.

Reviewer #2: Thank you for the detailed response letter and the comprehensive answers to the inquiries. I look forward to seeing this paper published.

7. PLOS authors have the option to publish the peer review history of their article (what does this mean?). If published, this will include your full peer review and any attached files.

Reviewer #1: No

Reviewer #2: No

---

## [Editor Report · Acceptance letter]

30 Oct 2024

PONE-D-24-20764R1 

PLOS ONE

Dear Dr. Burghardt, 

I'm pleased to inform you that your manuscript has been deemed suitable for publication in PLOS ONE. Congratulations! Your manuscript is now being handed over to our production team.

Kind regards, 

on behalf of

Dr. Stefano Cresci 

Academic Editor

PLOS ONE